# OpenReview forum: "Your Large Reasoning Models Can Be Safer on Its Own"
_ICLR.cc/2026/Conference — ICLR 2026 Conference Withdrawn Submission_

### Official Review · Reviewer_rmPS · 2025-10-26

**Soundness:** 2
**Presentation:** 3
**Contribution:** 2
**Rating:** 2
**Confidence:** 4

**Summary:**

This paper reveals that Large Reasoning Models (LRMs) possess a "Latent Safety Awareness," an inherent ability to identify safety risks in their own reasoning when prompted to review both the query and their reasoning path. To activate this capability, the authors propose the "Safe Trigger" approach, a structured mechanism that inserts a safety analysis step before the final answer generation for risky queries. The method uses Supervised Fine-Tuning (SFT) to teach the model when to trigger this analysis and Direct Preference Optimization (DPO) to enhance the quality of the safety guidance. Experimental results show this approach significantly improves safety alignment against harmful and jailbreaking queries with almost no impact on general performance. The entire training process is self-sufficient, relying only on the model's own capabilities without needing manual annotation or more powerful external models.

**Strengths:**

1. The proposed "Safe Trigger" approach is novel, well-motivated, and significantly boosts the LRM's safety alignment, improving its defense against both direct harmful queries and jailbreaking attacks. The experiment shows that both the SFT and DPO stages improve safety performance.
2. The safety enhancements are achieved with almost no negative impact on the model's general performance, reasoning abilities, or user experience, and they also avoid over-refusing safe queries.
3. The entire training process is self-sufficient, relying solely on the model's own generation and reasoning capabilities. This eliminates the need for expensive manual annotation or reliance on more powerful closed-source models, making it a low-cost and scalable.

**Weaknesses:**

1. The finding of "Latent Safety Awareness" of LRMs (Section 2), where models can identify safety issues in their own reasoning when prompted to review both the query and their reasoning path, is already identified and verified in prior work [1]. Yet the submission did not cite or discuss it.
2. Prior LRM safety alignment methods are not compared [1, 2].
3. There seems to be a huge safety result difference on the StrongReject and WildJailbreak dataset compared to the numbers reported in the Star-1 paper, with the same llama-guard safety evaluator, which makes the comparison result not very convincing.
3. The implementation detail for the Star1 baseline is unclear; is it the same as the SFT setting as "Safe Trigger SFT"?
4. There should be a discussion on the effectiveness of the safety evaluation method in the paper.

[1] SafeKey: Amplifying Aha-Moment Insights for Safety Reasoning

[2] How Should We Enhance the Safety of Large Reasoning Models: An Empirical Study

**Questions:**

1. How does the proposed SFT method apply to the Star-1 training dataset? It will be beneficial to demonstrate that the proposed SFT response structure is effective for different training data.

---

### Official Review · Reviewer_bqg2 · 2025-10-31

**Soundness:** 3
**Presentation:** 3
**Contribution:** 3
**Rating:** 4
**Confidence:** 3

**Summary:**

This paper introduces the concept of "Latent Safety Awareness" in large reasoning models, positing that these models can identify safety risks in their own reasoning processes when prompted to reflect on both the query and their intermediate thoughts. Building on this observation, the authors propose "Safe Trigger," a two-stage training method (SFT followed by DPO) that inserts a structured safety analysis module between reasoning and final answer generation for risky queries. The approach uses self-generated training data from the target models, avoiding external annotations or stronger models. Experiments claim substantial safety improvements with negligible impact on general performance.

**Strengths:**

1. The idea of leveraging self-reflection for safety is intuitive and aligns with recent trends in chain-of-thought prompting and alignment techniques. The "Safe Trigger" mechanism is a clean, modular intervention that could inspire lightweight safety enhancements.

2. The self-supervised data construction is a practical contribution, reducing reliance on costly human labeling or proprietary APIs, which could appeal to resource-constrained researchers.

3. Over-refusal analysis suggests Safe Trigger does not substantially increase false refusals; some settings even reduce them.

**Weaknesses:**

1. The core idea of inserting an intermediate safety analysis before the final answer conceptually overlaps with a significant body of prior work on reflection, self-critique, and safety-aware reasoning. While the paper frames the approach as specific to LRMs, the mechanism itself resembles established "analyze-then-answer" safety templates. The paper must better articulate its novel technical contribution beyond the implementation details of the <safe> tag and the specific DPO reward function.

2. The SFT data are generated by the same model family; the DPO reward uses model-autograded binary checks themselves judged by the original model. This circularity invites reward hacking and miscalibration: the policy is optimized to satisfy its own (or a closely related) judge rather than ground-truth safety. At minimum, authors should (i) swap in out-of-distribution judges for reward filtering and (ii) show cross-model generalization of the reward.

3. This baseline removes the "safe" module but uses the same training data structure. The large performance gap suggests that the gains may stem from the architectural intervention of the "safe" module itself, rather than from the quality of the learned safety reasoning. A more informative control would integrate the safety analysis into a standard Chain-of-Thought format within the "think" block to test if the explicit, separate module is truly necessary.

4. Weak baselines: The comparisons are limited to underpowered methods. STAR uses only 1k curated samples, System Prompt is a simple zero-shot prefix, and No Trigger SFT is a weak ablation.

**Questions:**

Could you provide more qualitative analysis of some failure cases? For instance, examples of risky queries where the Safe Trigger was not activated, or safe queries where it was incorrectly activated.

---

### Official Review · Reviewer_TjbT · 2025-11-03

**Soundness:** 3
**Presentation:** 3
**Contribution:** 2
**Rating:** 4
**Confidence:** 3

**Summary:**

The paper introduces Safe Trigger, a structured safety module inserted between a model’s reasoning and final answer to activate LRMs’ “Latent Safety Awareness,” trained via SFT then DPO using only model-generated data (no manual labels/closed models). It markedly boosts Safety Alignment Rates (e.g., DeepSeek-8B on WildJailbreak to 82.25%) while minimally affecting general capabilities and triggering reliably on risky queries.

**Strengths:**

A “Safe Trigger” module inserts an explicit <safe>…</safe> analysis between reasoning and the final answer, with a well-defined, interpretable structure (core-issue risk, reasoning-process risk, guidance) and is only activated on risky/uncertain inputs.

Self-contained training pipeline. SFT on a 30k structured dataset (10k each general/harmful/jailbreak) followed by DPO on 20k WildJailbreak queries using a model-judged reward over Fsafe/Sexist/Tfull/Sfull—no manual labels or closed models required.

Generalizable and reproducible. Demonstrated on two distinct LRMs (Qwen3-8B, DeepSeek-R1-Distill-Llama-8B); prompts, training, and evaluation code are provided for replication.

**Weaknesses:**

(1) My main concern is the incremental novelty compared to existing safety alignment works in LRM as you listed in the related works, could you highlight more about what is the difference with previous safe LRM?

(2) For the rewarding labeling part, what is the motivation of using original model as the judge instead of a more powerful model? what is the motivation for the design of eq (1)? How many preference pairs are selected after filtering from 20k?

(3) It would be good to also include the baseline methods' performance regarding the general capability as the optimal method should balance the security and performance drop. Also would be good to include more baseline methods besides STAR1.

**Questions:**

will the curated structured /preference training dataset be released?

---

### Official Review · Reviewer_EDXq · 2025-11-07

**Soundness:** 3
**Presentation:** 3
**Contribution:** 2
**Rating:** 4
**Confidence:** 3

**Summary:**

They propose a new safety alignment method, Safe Trigger. They first reveal that LRMs have the Latent Safety Awareness. That is, when the LRMs can simultaneously see both the original harmful queries and their own original reasoning path, they can successfully recognize their own reasoning vulnerabilities and refuse to continue generating potentially harmful answers. They test it with SFT and DPO and compare their methods with other existing methods. Moreover, they show that their method doesn't degrade general capabilities.

**Strengths:**

- They first reveal that LRMs exhibit Latent Safety Awareness.
- They propose a new safety alignment method, Safe Trigger, motivated by this inherent Latent Safety Awareness in LRMs.
- They evaluate their method using two models and compare Safe Trigger with other approaches, showing that it outperforms existing methods.
- They also demonstrate that their method doesn't compromise the models’ general capabilities.

**Weaknesses:**

- Their method relies on an LLM judge, Llama-Guard-3-8B, to evaluate the safety alignment of models. This judge is also used to construct the dataset for Safe Trigger. However, since the judge may introduce some biases (especially when the variant of Llama 8B evaluates the safety of another variant of Llama 8B, that is, DeepSeek-R1-Distill-Llama-8B), they need to assess the reliability of the LLM judge through some annotation.
- They evaluate only two models, which reduces the generalizability of the method.
- In fact, looking at Table 3, the effectiveness of Safe Trigger seems limited. For example, Qwen3-8B (a more capable model) shows a smaller performance gap between Safe Trigger and other methods compared to DeepSeek-R1-Distill-Llama-8B. Will the effectiveness of Safe Trigger decrease as the model becomes more capable?
- I think it would also be useful to examine whether the effectiveness of Safe Trigger increases when the dataset is constructed using other models instead of the target model itself.

**Questions:**

Please see the weaknesses

---

### Note · Authors · 2025-12-02

I have read and agree with the venue's withdrawal policy on behalf of myself and my co-authors.